### Bowen ratio-constrained global dataset of air-sea turbulent heat 1 2 fluxes from 1993 to 2017 Yizhe Wang<sup>a, b</sup>, Ronglin Tang<sup>a, b, \*</sup>, Meng Liu<sup>c</sup>, Lingxiao Huang<sup>a, b</sup>, Zhao-Liang Li<sup>a, b, c</sup> 3 4 <sup>a</sup> State Key Laboratory of Resources and Environment Information System, Institute of 5 Geographic Sciences and Natural Resources Research, Chinese Academy of Sciences, 6 Beijing 100101, China <sup>b</sup> University of Chinese Academy of Sciences, Beijing 100049, China 7 8 <sup>c</sup> State Key Laboratory of Efficient Utilization of Arable Land in China, Institute of 9 Agricultural Resources and Regional Planning, Chinese Academy of Agricultural 10 Sciences, Beijing 100081, China \* Authors to whom correspondence should be addressed: tangrl@lreis.ac.cn 11 12 13 14

### 15 Abstract

16 Air-sea turbulent heat fluxes, including the sensible heat flux (SHF) and latent heat 17 flux (LHF), along with the Bowen ratio ( $\beta$ , ratio of SHF to LHF), are crucial for 18 understanding air-sea interaction and global energy and water budgets. However, the 19 existing products, primarily developed using the semi-empirical bulk aerodynamic 20 methods and data-driven machine learning approaches, are often weak in accuracy and 21 physical rationality, due to the uncertainties in the environmental forcings and 22 inappropriate parameterizations. In this study, we generated a global daily 0.25° product 23 of air-sea turbulent heat fluxes using the Bowen ratio-constrained Neural Network (NN) 24 model (referred to as the BrTHF model) that could coordinately estimate the SHF and 25 LHF, along with the observations from 197 globally distributed buoys and multi-source 26 remote sensing and reanalysis forcings. The spatial ten-fold cross-validation results 27 showed that the BrTHF model, achieving root mean square errors of 6.05 W/m<sup>2</sup>, 23.67 28 W/m<sup>2</sup> and 0.22 and correlation coefficients of 0.93, 0.91 and 0.25 for the SHF, LHF and 29  $\beta$ , respectively, outperformed the physics-agnostic NN model and seven widely used 30 air-sea turbulent heat flux products (including JOFURO3, IFREMER, SeaFlux, ERA5, 31 MERRA2, OAFlux, and OHF). Furthermore, the inter-comparison of the spatial 32 distribution of multi-year means, as well as intra-annual and inter-annual change 33 patterns showed that the BrTHF product reliably simulated global SHF, LHF and  $\beta$ , in 34 contrast to the machine learning-based OHF product that failed to replicate these 35 patterns. The main advantage of the BrTHF model lies in its improved rationality of  $\beta$ 36 estimates, successfully eliminating the outliers observed in the physics-agnostic NN 37 model and the seven typical products. The improved SHF, LHF, and  $\beta$  estimates can 38 allow for more accurate quantification of the global air-sea energy and water budgets, 39 enhance our understanding of air-sea interaction, and improve projections of climate 40 change under global warming. The 0.25° daily global product from 1993 to 2017 can 41 be freely accessed from the National Tibetan Plateau Data Center (TPDC) 42 [https://doi.org/10.11888/Atmos.tpdc.302578, Tang and Wang (2025)].

Keywords: Air-sea turbulent heat fluxes; Sensible heat flux; Latent heat flux; Bowen

- ratio
- **1. Introduction**

Air-sea turbulent heat fluxes (THF), comprising the evaporative latent heat flux 47 (LHF) and conductive sensible heat flux (SHF), play vital roles in the Earth's climate 48 system by characterizing the exchange of energy and water between the ocean and 49 atmosphere (Wild et al., 2014; Loeb et al., 2021; Fasullo et al., 2014). The ratio, 50 commonly referred to as the Bowen Ratio ( $\beta = SHF/LHF$ ), serves as a key indicator 51 revealing the partitioning of water and energy over the ocean and atmosphere (Jo, 2002; 52 Andreas et al., 2013; Liu and Yang, 2021). Accurate estimation of these three 53 parameters is an essential prerequisite for advancing our understanding of 54 atmosphere-sea interaction (Gentemann et al., 2020), improving the quantification of 55 global water and energy budget (Zhang, 2023), and enhancing the predictability of 56 extreme weather events (Yu, 2019).

To map global air-sea turbulent heat fluxes, the semi-empirical bulk aerodynamic 58 method, which establishes scaling relationships between flux and profiles of easily 59 measured mean metrological quantities, such as near-surface gradients of humidity, 60 temperature and wind (Yu, 2019), based on the Monin-Obukhov similarity theory (Monin and Obukhov, 1954), was developed and widely adopted as a primary approach. 61 62 This method, for its ease of application, has been applied to generate tens of widely 63 used products in the past few decades (Shie et al., 2009; Liman et al., 2018; Yu and Weller, 2007; Berry and Kent, 2011; Tomita et al., 2018; Crespo et al., 2019). However, 64 65 there were huge discrepancies in the global and regional magnitude and patterns of SHF and LHF among these products, which seriously imped our understanding of the key 66 67 process of the air-sea interaction and the global budget of water and energy (Bentamy 68 et al., 2017; Tang et al., 2024; Yu, 2019). The discrepancies could be partly ascribed to 69 the substantial uncertainties in the environmental forcings used to develop these 70 products (Robertson et al., 2020) and the inappropriate parameterizations regarding

regional atmospheric stability and boundary layer dynamics, across diverse and 72 complex environmental conditions (Brodeau et al., 2017; Jiang et al., 2024a; Jiang et 73 al., 2024b; Yang et al., 2024). Furthermore, these problems contribute a lot to the biases 74 in the SHF and LHF estimates which can even lead to the unphysical estimations of  $\beta$ , 75 as Wang et al. (2024) reported. To better describe and comprehend the air-sea 76 interaction and the energy and water budgets, the existing mode to produce global air-77 sea turbulent heat fluxes needs improvement urgently.

Machine learning techniques have been extensively applied in up-scaling in situ 79 measurements of a single variable (e.g. soil moisture, roughness or temperature) to the 80 globe (Wang et al., 2023; Peng et al., 2022; O and Orth, 2021; Nelson et al., 2024; Fu 81 et al., 2023). These efforts highlight the great potential of machine learning for more 82 accurate and consistent multivariate coordinated mapping (Karniadakis et al., 2021; 83 Kashinath et al., 2021; Van Der Westhuizen et al., 2023; Wang et al., 2024). However, 84 the application of machine learning in global mapping of air-sea turbulent heat fluxes 85 remains limited. The only publicly available machine learning-based global air-sea turbulent heat fluxes product, released by the National Oceanic and Atmospheric 86 87 Administration (NOAA) Ocean heat flux CDR (hereafter dubbed OHF), 88 simultaneously modeled SHF and LHF using a Neural Network (NN) technique 89 (Clayson and Brown, 2016). Although it performed well when validated against the 90 observations from the tropical buoys, it failed to capture the regional characteristics, 91 particularly in areas where air-sea turbulent heat exchange is intense (e.g. oceans with 92 latitudes beyond 45° for SHF and subtropical highs for LHF) (Tang et al., 2024). Additionally, it exhibited different pattern of temporal evolution of global annual mean 93 94 and opposite inter-annual trends at both regional and global scales to most widely-used 95 physical model-based products, likely due to unreasonable construction of observation 96 datasets [with data before and after 2007 coming from SeaFlux in-situ datasets and 97 ICOADS (International Comprehensive Ocean-Atmosphere Data Set) datasets, 98 respectively]. Furthermore, the product likely suffers from unphysical estimates of the

$\beta$  due to neglecting the interrelations among SHF, LHF and  $\beta$  during the model 100 construction.

To improve the estimation of SHF, LHF, and  $\beta$  in a coordinative framework, we 102 recently proposed an innovative Bowen ratio-informed data-driven model by 103 considering their synergistic changes using a Random Forest (RF) technique (Wang et al., 2024). Validation against hourly high-quality eddy covariance (EC) flux 104 105 measurements from 53 historical cruises demonstrated the model's superior 106 performance, achieving high accuracy in estimating SHF, LHF, and  $\beta$ , with results that 107 are physically consistent. This work highlights the feasibility of simultaneously 108 estimating SHF, LHF, and  $\beta$  with high accuracy using machine learning techniques, 109 offering strong potential for global mapping that aligns with physical consistency. 110 However, due to limited availability of EC flux measurements (characterized by sparse spatio-temporal distributions), the application of the model for global mapping remains 111 112 constrained. Buoy-based flux observations provide a viable alternative. Although less 113 reliable than EC-based flux measurements, buoy data offer globally representative flux 114 samples with adequate volume and acceptable accuracy, which have been widely used 115 to evaluate the performance of global products (Bentamy et al., 2017; Tang et al., 2024; 116 Weller et al., 2022; Zhou et al., 2020) and support global modeling (Chen et al., 2020a) 117 and analysis (Song et al., 2024; Yan et al., 2024).

The primary objectives of this study are three-folds: (1) to develop an innovative 119 Bowen ratio-constrained model for improving the air-sea SHF, LHF and  $\beta$  estimates 120 (referred to as the BrTHF model hereafter) using the machine learning technique and 121 global buoy-based air-sea turbulent heat fluxes observations; (2) to demonstrate the 122 superiority of the model through an inter-comparison with seven widely used global 123 products and the estimates from the physics-free machine learning-based model; (3) to 124 produce a global daily 0.25° dataset based on the BrTHF model over ice-free oceans 125 covering the period from 1993 to 2017. The flux observations from 197 global 126 distributed buoys, along with multi-source satellite-based and reanalysis-based forcings,

- were collected to construct the models and further produce the global air-sea turbulent
- heat fluxes dataset. The accuracy and spatio-temporal patterns of the SHF, LHF and  $\beta$
- estimates were inter-compared with seven widely used products, including the remote
- sensing-based JOFURO v3, IFREMER v4.1 and SeaFlux v3, as well as reanalysis-
- based ERA5 and MERRA2, hybrid-based OAFlux v3 and machine learning-based
- OHF v2 products.
- **2. Data and Methods**
- The following sub-sections provide an overview of the development of the BrTHF
- product, detailing the construction of air-sea turbulent heat fluxes observation datasets,
- forcing datasets and the BrTHF model, as well as the evaluation strategies used in this
- study, as indicated in Figure 1.