# Peer review of "Bowen ratio-constrained global dataset of air-sea turbulent heat 1 2 fluxes from 1993 to 2017 Yizhe Wanga, b, Ronglin Tanga, b, \*, Meng Liuc, Lingxiao Huanga, b, Zhao-Liang Lia, b, c 3 4 a State Key Laboratory of Resources and En"

_Earth System Science Data, 2025_

## Referee Comment (RC2)

Review of 'Bowen ratio-constrained global dataset of air-sea turbulent heat fluxes from 1993 to 2017'

By Yizhe Wang, Ronglin Tang, Meng Liu, Lingxiao Huang, and Zhao-Liang Li

The authors produced a heat flux dataset based on a statistical neural network trained over model reanalyses and / or buoy data (I am not sure, it is not so clear to me after reading their manuscript). They compare their product to other products, and mostly find that their product performs better.

There are strong chances that conceptually, this whole work would be no use, since the reanalyses used to train their network already provide the surface fluxes. Therefore, I really don't see the point in producing what I would call a 'statistical shortcut' of an existing model.

My interpretation of the context is that historically, global surface flux datasets were developed at a time when model reanalyses were not accurate enough. In this context, independent blended-analyses gathering various satellite sensor fields and sometimes model forecasts (for stability and / or near surface air temperature) could be helpful for documenting the heat budget and it is spatial variability. Nowadays, satellite sensor data as well as *in situ* observations are widely assimilated in models, which results -in my opinion- in an optimum mix between physics (equations in the models) and observations, in terms of surface heat fluxes. Therefore, I don't see why independent flux products (which are not even an ounce independent from models, since they are trained on them) should be developed any longer, the reason for which I left this field. At best, the authors' product will perform the same as model fields, which is obvious according to Figure 3 and 4 (compare ERA in panels d, to 'BrTHF' in panel i). Worse, there is one risk when aiming at getting the highest accuracy with artificial neural networks: overtraining. This could have been discussed in the manuscript.

Please note that the proposed BrTHF product does not account for negative LHF values (Figure 4i)

The authors focus on the Bowen ratio, which is supposed to give more 'consistency in physics', I don't even know how to define/name it as they do... I am not convinced at all. Technically, I think it is just a matter of optimizing their neural network configuration.

To me, it seems that the authors have downloaded a lot of data and model fields, and that they desperately look for a way to add some value using these datasets. If so, I would rather encourage the authors to analyze what is inside and produce case analyses, statistical analyses.

In this manuscript, the principal publications are not even cited, which I consider to be a lack of respect to authors that did a pioneering work more than twenty years before them!

- Bourras, D., L. Eymard, & Liu, W. T. (2002). A neural network to estimate the latent heat flux over oceans from satellite observations, International Journal of Remote Sensing, 23(12), 2405-2423. doi: http://doi.org/10.1080/01431160110070825
- Bourras, D., Liu, W. T., Eymard, L., & Tang, W. (2003). Evaluation of Latent Heat Flux Fields from Satellites and Models during SEMAPHORE, Journal of Applied Meteorology, 42(2), 227-239. doi: https://doi.org/10.1175/1520-0450(2003)042<0227:EOLHFF>2.0.CO;2
- Bourras, D. (2006). Comparison of Five Satellite-Derived Latent Heat Flux Products to Moored Buoy Data, Journal of Climate, 19(24), 6291-6313. doi: https://doi.org/10.1175/JCLI3977.1
- Bourras, D., Reverdin, G., Caniaux, G., & Belamari, S. (2007). A Nonlinear Statistical Model of Turbulent Air–Sea Fluxes, Monthly Weather Review, 135(3), 1077-1089. doi: https://doi.org/10.1175/MWR3335.1

Some comments for the introduction:

-L46 'the evaporative latent heat flux': the term 'evaporative' is not appropriate in this sentence

-L47 'the conductive sensible heat flux': wrong, it is convection, not conduction, except in the first microns above the water surface

-L51 'the Bowen ratio…revealing the partitioning of water and energy over the ocean and atmosphere': this sentence does not make any sense, and it is not helpful, in addition to what the definition of the Bowen ratio is common knowledge in this field

-L52-L54: 'Accurate estimation of these three parameters is an essential prerequisite for advancing our understanding of atmosphere-sea interaction'… I don't see why the Bowen ratio would be key, and the fluxes as well as the Bowen ratio are not 'parameters' but 'variables', in this context

-L57-L61: 'To map global air-sea... as developed and widely adopted as a primary approach'. This sentence is nonsense. The Monin-Obukhov (1954) similarity theory was not developed for that, and I am not aware of any 'primary approach'

-L58: 'easily': I don't see why it would be 'easy' to measure mean meteorological quantities, it is rather complicated, just try to get a reliable information with two thermometers mounted close to each other on a ship or on a buoy, it is a real challenge. In addition, this includes SST, which is not a meteorological variable, strictly speaking

-L59: 'metrological': Wrong, I think the authors mean 'meteorological'

After reading this one and half paragraph I have noted so many inaccuracies and / or wrong statements, that I don't feel compelled to review in detail the rest of the manuscript. This manuscript looks like a science paper, but from far. To me, it is way too weak to be published.

Other comments, maybe not in order of line numbering:

-The manuscript is unnecessarily long, difficult to read. It contains unnecessary acronyms such as THF, and it contains unnecessary equations, such as the equation 1 that relates the relative humidity to the dew point temperature, which is common knowledge

-Figure 1 is unclear

-Figure 5 is statistically pointless

-At several locations in the manuscript, the terminology used may be considered as misleading, such as L122 where they mention 'the superiority of the model'. In this sentence, 'model' is ambiguous because it does not refer to a meteorological model or a physical model of any kind, but rather to a statistical model. At L136, there is also a reference to the 'BrTHF model'

-In the same fashion, section 2.2 is entitled 'forcing datasets', which I think also adds to the confusion, because forcing is usually used by ocean modelers. Here, it should be 'learning', which term is widely used in the field of multilayer perceptrons

-In section 2.2, I could not easily understand whether only model analyses were used for the learning (which I think), or if it is a mix with buoy data

-L112-L113 and L104: the authors mention several times the EC fluxes are high quality compared to bulk estimates, which denotes a complete lack of knowledge in this field. EC fluxes are very difficult to obtain at sea because of platform motion and airflow distortion, even at turbulent scales. To get more insights, the authours should consider reading the following references, for example:

- Bourras, D., Weill, A., Caniaux, G., Eymard, L., Bourlès, B., Letourneur, S., Legain, D., Key, E., Baudin, F., Piguet, Traullé, O., Bouhours, G., Sinardet, G., Barrié, J., Vinson, J.-P., Boutet, F., Berthod, C., &

Clémençon, A. (2009). Turbulent air-sea fluxes in the Gulf of Guinea during the AMMA Experiment, *J. Geophys. Res.*, 114, C04014. doi: https://doi.org/10.1029/2008JC004951

- Bourras, D., Cambra, R., Marié, L., Bouin, M.-N., Baggio, L., Branger, Beghoura, H., Reverdin, G., Dewitte, B., Paulmier, A., Maes, C., Ardhuin, F., Pairaud, I., Fraunié, P., Luneau, C., & Hauser, D. (2019). Air-sea turbulent fluxes from a wave-following platform during six experiments at sea, *J. Geophys. Res.*, 124, 4290–4321. doi: https://doi.org/10.1029/2018JC014803

---

## Author Comment (AC2)

**Responses to the Comments and Suggestions**

**Reviewer 2:**

The authors produced a heat flux dataset based on a statistical neural network trained over model reanalyses and / or buoy data (I am not sure, it is not so clear to me after reading their manuscript). They compare their product to other products, and mostly find that their product performs better.

Re: Thank you for your comments. We have carefully considered all your comments and suggestions and made corresponding point-by-point responses and revisions. Specifically, reviewer comments are shown in black, our responses in blue, and the corresponding revisions in the manuscript are highlighted in red. We hope that our responses and the revised manuscript would be satisfactory.

There are strong chances that conceptually, this whole work would be no use, since the reanalyses used to train their network already provide the surface fluxes. Therefore, I really don't see the point in producing what I would call a 'statistical shortcut' of an existing model.

Re: Thank you for your comment. We would like to clarify that the target variables in our study are sourced from in-situ buoy observations, rather than from outputs provided by reanalysis products. Although the input features for our model include variables from reanalysis, the neural network is trained to reproduce observed air-sea turbulent heat fluxes, rather than merely replicating outputs of reanalysis. Accordingly, our approach should not be considered a "statistical shortcut" of existing models, but rather a methodology aimed at improving air-sea turbulent heat fluxes estimation by integrating observations with machine learning techniques.

My interpretation of the context is that historically, global surface flux datasets were developed at a time when model reanalyses were not accurate enough. In this context, independent blended-analyses gathering various satellite sensor fields and sometimes model forecasts (for stability and / or near surface air temperature) could be helpful for documenting the heat budget and it is spatial variability. Nowadays, satellite sensor data as well as in situ observations are widely assimilated in models, which results -in my opinion- in an optimum mix between physics (equations in the models) and observations, in terms of surface heat fluxes. Therefore, I don't see why independent flux products (which are not even an ounce independent from models, since they are trained on them) should be developed any longer, the reason for which I left this field.

Re: Thank you for your comment. We respectfully note that we do not fully agree with the reviewer's perspective. Currently, multiple global reanalysis products exist, and these products are still under development and not fully mature, which contrasts with the implication that additional independent flux products are unnecessary and that reanalysis represents an optimal mix between physics and observations. While we acknowledge that assimilating satellite data and in-situ observations into process-based models can improve the accuracy of air–sea turbulent heat fluxes simulations, it should be recognized that the accuracy of flux estimates is significantly influenced by the physical representation of air–sea exchange processes, model parameterizations, and biases in inputs. Therefore, assimilation alone does not necessarily guarantee high-accuracy flux estimates, which partially explains the continued need for model development and optimization.

With the rapid growth in the availability of flux observations, integrating machine learning models while fully accounting for the key physical and environmental factors influencing air–sea turbulent heat exchange has become an important approach for improving the accuracy and reliability of air–sea turbulent heat fluxes estimations. Indeed, in recent years, global estimations of carbon, water, and energy fluxes, ocean currents, and temperature/salinity fields using machine learning trained on in situ observations have become increasingly common (Chen et al., 2019; Cummins et al., 2023; Cutolo et al., 2024; Ge et al.; Zhou et al., 2024). Moreover, AI-driven models such as AlphaFold have achieved breakthrough progress in protein structure prediction, illustrating the substantial potential of artificial intelligence (Jumper et al., 2021). Other notable examples include OpenAI's GPT series in natural language understanding and generation, DeepMind's AlphaZero in surpassing human performance in complex strategy games, and deep-learning–based climate model parameterization and Earth system prediction (Brown et al., 2020; Rasp et al., 2018; Silver et al., 2018). Collectively, these successes demonstrate that scientific research is increasingly embracing a "data-driven + AI-assisted". In our view, flux estimation should be continuously improved by integrating emerging technologies in order to provide more accurate and reliable results.

At best, the authors' product will perform the same as model fields, which is obvious according to Figure 3 and 4 (compare ERA in panels d, to 'BrTHF' in panel i). Worse, there is one risk when aiming at getting the highest accuracy with artificial neural networks: overtraining. This could have been discussed in the manuscript. Please note that the proposed BrTHF product does not account for negative LHF values (Figure 4i).

Re: Thank you for your comment. With regard to the concern that our product may not outperform reanalysis, Figures 3 and 4 show that BrTHF achieves substantial improvements over ERA5, with RMSE reductions of ~1 W/m$^2$ (14%) for SHF and ~5 W/m$^2$ (16%) for LHF.

To address the reviewer's concern about potential overfitting, we implemented two measures to ensure the robustness and generalizability of our model. First, we employed a spatial 10-fold cross-validation, which provides a rigorous evaluation of model performance. Second, following the suggestions of another reviewer, we conducted targeted cross-validation by withholding two high-latitude buoy sites in the Southern Hemisphere, largely independent from the training dataset, as a validation set. As shown in Tables S4 and S5, BrTHF maintained higher accuracy than the other products and models, demonstrating its reliable generalization ability.

Regarding negative LHF values, we note that small negative values remain in Figure 4

(i), but their magnitudes are close to zero. This is mainly due to the uneven distribution of observations and the constraints applied to the BrTHF model, which prioritizes simultaneous high-accuracy estimation of SHF, LHF, and $\beta$. Consequently, the predicted range is compressed. We acknowledge this limitation and have discussed it in Section 3.5 of the original manuscript.

The authors focus on the Bowen ratio, which is supposed to give more 'consistency in physics', I don't even know how to define/name it as they do… I am not convinced at all. Technically, I think it is just a matter of optimizing their neural network configuration.

Re: Thank you for your comment. Regarding "consistency in physics," we would like to clarify that our main goal is to ensure that model outputs satisfy the physical relationship SHF/LHF = $\beta$. While this relationship can indeed be maintained in the reanalysis products highlighted by the reviewer, as shown in Figures 3–6, these reanalysis models cannot simultaneously provide SHF, LHF, and $\beta$ with high accuracy.

Conversely, using machine learning to model SHF, LHF, and $\beta$ separately can achieve high accuracy for each variable individually, but such predictions do not necessarily preserve the physical relationship SHF/LHF = $\beta$. Therefore, our work emphasizes achieving both physical consistency (SHF/LHF = $\beta$) and high-accuracy estimation of all three variables, which demonstrates that our approach is not merely an optimization of the neural network configuration.

To me, it seems that the authors have downloaded a lot of data and model fields, and that they desperately look for a way to add some value using these datasets. If so, I

would rather encourage the authors to analyze what is inside and produce case analyses, statistical analyses.

Re: Thank you for your comment. We would like to clarify that our study is not a simple aggregation of existing data. Instead, it aims to improve simultaneous estimation of

SHF, LHF, and $\beta$ through a physically-informed neural network—a novel approach beyond existing case studies or statistical analyses. The results demonstrate that BrTHF

reduces both RMSE and bias of SHF and LHF compared to the existing state-of-the-art products. We believe this constitutes a meaningful contribution to the ongoing efforts in improving air–sea turbulent heat fluxes estimation.

In this manuscript, the principal publications are not even cited, which I consider to be a lack of respect to authors that did a pioneering work more than twenty years before them!

● Bourras, D., L. Eymard, & Liu, W. T. (2002). A neural network to estimate the latent heat flux over oceans from satellite observations, International Journal of

Remote           Sensing,           23(12),           2405-2423.           doi:

http://doi.org/10.1080/01431160110070825

● Bourras, D., Liu, W. T., Eymard, L., & Tang, W. (2003). Evaluation of Latent Heat

Flux Fields from Satellites and Models during SEMAPHORE, Journal of Applied

Meteorology,       42(2),       227-239.       doi:       https://doi.org/10.1175/1520-

0450(2003)0422.0.CO;2

● Bourras, D. (2006). Comparison of Five Satellite-Derived Latent Heat Flux

Products to Moored Buoy Data, Journal of Climate, 19(24), 6291-6313. doi:

https://doi.org/10.1175/JCLI3977.1

● Bourras, D., Reverdin, G., Caniaux, G., & Belamari, S. (2007). A Nonlinear

Statistical Model of Turbulent Fluxes, Monthly Weather Review, 135(3), 1077-

1089. doi: https://doi.org/10.1175/MWR3335.1

Re: Thank you for your comment. We fully acknowledge and respect the contributions of the pioneering studies, and in the revised manuscript, we have now carefully revised the manuscript to include appropriate citations to these important references. We thank the reviewer for pointing this out.

Some comments for the introduction:

-L46 'the evaporative latent heat flux': the term 'evaporative' is not appropriate in this sentence

Re: Thank you for your comment. We have removed the redundant term "evaporative" and now simply use "latent heat flux" for clarity.

-L47 'the conductive sensible heat flux': wrong, it is convection, not conduction, except in the first microns above the water surface

Re: Thank you for your comment. We sincerely apologize for the typo and have corrected the relevant description accordingly.

-L51 'the Bowen ratio…revealing the partitioning of water and energy over the ocean and atmosphere': this sentence does not make any sense, and it is not helpful, in addition to what the definition of the Bowen ratio is common knowledge in this field

Re: Thank you for your comment. In the revised manuscript, we have removed the related description and now provide the definition of the $\beta$ upon its first appearance.

-L52-L54: 'Accurate estimation of these three parameters is an essential prerequisite for advancing our understanding of atmosphere-sea interaction'… I don't see why the Bowen ratio would be key, and the fluxes as well as the Bowen ratio are not 'parameters' but 'variables', in this context

Re: Thank you for your comment. We agree that the use of the term "parameters" in this context could be misleading, and we have revised it to "SHF, LHF and their ratio—

the Bowen ratio ($\beta$ = SHF/LHF)". We also acknowledge the reviewer's concern regarding the role of $\beta$. We would like to clarify that while SHF and LHF individually describe the components of turbulent heat fluxes, $\beta$ provides additional insight into their relative partitioning at the air–sea interface. This ratio not only captures differences in climate regimes (e.g., large $\beta$ in cold and dry regions such as the subpolar North Atlantic, and small $\beta$ in tropical and subtropical oceans), but also reflects the synergistic variations between SHF and LHF (e.g., both SHF and LHF may increase while $\beta$

remains unchanged), which cannot be inferred from either flux alone. Therefore, we consider $\beta$ to be an essential variable for advancing the understanding of atmosphere–

ocean interactions.

-L57-L61: 'To map global air-sea... as developed and widely adopted as a primary approach'. This sentence is nonsense. The Monin-Obukhov (1954) similarity theory was not developed for that, and I am not aware of any 'primary approach'

Re: Thank you for your comment. We agree that the Monin–Obukhov similarity theory was not originally developed for mapping global air–sea fluxes, and it is not accurate to describe it as a 'primary approach' in this context. In the revised manuscript, we have revised the sentence to more appropriately reflect its role as a theoretical foundation widely used in flux parameterization schemes in the second paragraph of Section 1 as follows:

"To estimate global air–sea turbulent heat fluxes, the semi-empirical bulk aerodynamic method was developed based on the Monin–Obukhov similarity theory (Monin and

Obukhov, 1954). It establishes scaling relationships between fluxes and near-surface meteorological variables such as wind speed, humidity, and temperature (Yu, 2019)."

-L58: 'easily': I don't see why it would be 'easy' to measure mean meteorological quantities, it is rather complicated, just try to get a reliable information with two thermometers mounted close to each other on a ship or on a buoy, it is a real challenge.

In addition, this includes SST, which is not a meteorological variable, strictly speaking

Re: Thank you for your comment. We apologize for the inappropriate wording and have made the corresponding corrections in the manuscript. We also acknowledge that including sea surface temperature (SST) in this context was misleading, and we have now corrected this accordingly.

-L59: 'metrological': Wrong, I think the authors mean 'meteorological'

Re: Thank you for your comment. We have revised "metrological" to "meteorological".

After reading this one and half paragraph I have noted so many inaccuracies and / or wrong statements, that I don't feel compelled to review in detail the rest of the manuscript. This manuscript looks like a science paper, but from far. To me, it is way too weak to be published.

Re: We sincerely appreciate the reviewer's feedback. We fully acknowledge the concerns raised regarding inaccurate statements in the manuscript, and have carefully considered all comments, undertaking substantial revisions to address these issues. At the same time, we have incorporated the constructive suggestions and comments provided by another reviewer, which have further enhanced the clarity, rigor, and overall quality of the manuscript. We believe that, following these revisions, the manuscript now presents meaningful and valuable scientific contributions.

Other comments, maybe not in order of line numbering:

-The manuscript is unnecessarily long, difficult to read. It contains unnecessary acronyms such as THF, and it contains unnecessary equations, such as the equation 1

that relates the relative humidity to the dew point temperature, which is common knowledge

Re: Thank you for your comment. In the revised manuscript, we have removed the unnecessary acronyms and equations for conciseness.

-Figure 1 is unclear

Re: Thank you for your comment. We have reorganized the flowchart to improve its readability, as shown below:

[Figure]

**Figure 1. flowchart of the generation of a global product of air-sea SHF, LHF and $\beta$ by the**

**BrTHF model**

-Figure 5 is statistically pointless

Re: Thank you for your comment. We would like to clarify that the main purpose of

Figure 5 is to present and compare the distribution of $\beta$ estimates from our model and other products against observations. The highlighting of outliers in the Figure 5 is intended to demonstrate that our model effectively avoids the outliers found in other models and products. Additionally, since each panel shows two modes (with and without outliers), to maintain the figure's clarity and avoid redundancy, detailed statistical information can be found in Table S2 and Figure 6. To prevent any misunderstanding, we have added an explanation in the caption of Figure 5 in the revised manuscript:

**"Figure 5. Same as Figure 3 but for $\beta$. The samples out of the ranges of observed $\beta$ (-5 $\leq \beta \leq$ 5)**

**were colored in blue, orange, green, red, purple, brown, pink and gray for JOFURO3,**

**IFREMER, SeaFlux, ERA5, MERRA2, OAFlux, OHF products and the physics-free NN**

**models, respectively. The corresponding statistical metrics can be found in Table S3 and**

**Figure 6."**

-At several locations in the manuscript, the terminology used may be considered as misleading, such as L122 where they mention 'the superiority of the model'. In this sentence, 'model' is ambiguous because it does not refer to a meteorological model or a physical model of any kind, but rather to a statistical model. At L136, there is also a reference to the 'BrTHF model'

Re: Thank you for your comment. We agree that the term "model" may be ambiguous without clarification. In our original manuscript, we referred to the BrTHF model as "a

Bowen ratio-constrained model using the machine learning technique," which implicitly indicates that it is a statistical model. However, to avoid potential ambiguity for readers, we have revised the first appearance of the BrTHF model to clearly state that it is a *"Bowen ratio-constrained statistical model using the machine learning*

*technique"*. We continue to use the term "BrTHF model" throughout the manuscript for readability. Additionally, we have revised the sentence to specify that we are referring to the statistical model developed in this study.

-In the same fashion, section 2.2 is entitled 'forcing datasets', which I think also adds to the confusion, because forcing is usually used by ocean modelers. Here, it should be

'learning', which term is widely used in the field of multilayer perceptrons

Re: Thank you for your comment. We have revised the title of Section 2.2 to "Learning datasets for training the neural network" and updated related terminology throughout the manuscript.

-In section 2.2, I could not easily understand whether only model analyses were used for the learning (which I think), or if it is a mix with buoy data.

Re: Thank you for your comment. We would like to clarify that in our neural network framework, model analyses were used as input features, while buoy-based SHF and

LHF observations served as the target variables for training. Accordingly, we have revised the relevant descriptions in the second paragraph of Section 2.2.1 to improve clarity as follows:

"Datasets of these learning variables used as input features for training the neural network were collected from multiple publicly available sources, as summarized in

Table 2 and were used as the input features for training the neural network."

-L112-L113 and L104: the authors mention several times the EC fluxes are high quality compared to bulk estimates, which denotes a complete lack of knowledge in this field.

EC fluxes are very difficult to obtain at sea because of platform motion and airflow distortion, even at turbulent scales. To get more insights, the authours should consider reading the following references, for example:

• Bourras, D., Weill, A., Caniaux, G., Eymard, L., Bourlès, B., Letourneur, S.,

Legain, D., Key, E., Baudin, F., Piguet, Traullé, O., Bouhours, G., Sinardet, G.,

Barrié, J., Vinson, J.-P., Boutet, F., Berthod, C., & Clémençon, A. (2009). Turbulent air-sea fluxes in the Gulf of Guinea during the AMMA Experiment, J. Geophys.

Res., 114, C04014. doi: https://doi.org/10.1029/2008JC004951

• Bourras, D., Cambra, R., Marié, L., Bouin, M.-N., Baggio, L., Branger, Beghoura,

H., Reverdin, G., Dewitte, B., Paulmier, A., Maes, C., Ardhuin, F., Pairaud, I.,

Fraunié, P., Luneau, C., & Hauser, D. (2019). Air-sea turbulent fluxes from a wave- following platform during six experiments at sea, J. Geophys. Res., 124, 4290–

4321. doi: https://doi.org/10.1029/2018JC014803

Re: Thank you for your comment. We would like to clarify that our reference to EC

fluxes as "high quality" was intended to emphasize their value as direct measurements of turbulent heat fluxes, rather than to suggest that they are easy to obtain. We fully acknowledge that EC measurements at sea are challenging due to platform motion and airflow distortion, even at turbulent scales. In the revised manuscript, to avoid possible misinterpretation, we have removed the wording describing EC fluxes as "high quality"

and have revised similar statements elsewhere in the manuscript. Furthermore, we have carefully reviewed the literature recommended by the reviewer and added these references in the fifth paragraph of Section 1 to highlight the challenges of obtaining

EC measurements over the ocean as follows:

"However, since EC observations are difficult to obtain at sea due to platform motion and airflow distortion (Bourras et al., 2019; Bourras et al., 2009)—their limited spatio- temporal coverage constrains the application of the model for global mapping."

Reference:

Bourras, D., Cambra, R., Marié, L., Bouin, M.N., Baggio, L., Branger, H., Beghoura,
H., Reverdin, G., Dewitte, B., Paulmier, A., Maes, C., Ardhuin, F., Pairaud, I.,
Fraunié, P., Luneau, C. and Hauser, D., 2019. Air‐Sea Turbulent Fluxes From
a Wave‐Following Platform During Six Experiments at Sea. Journal of
Geophysical Research: Oceans, 124(6): 4290-4321.

Bourras, D., Weill, A., Caniaux, G., Eymard, L., Bourlès, B., Letourneur, S., Legain,
D., Key, E., Baudin, F., Piguet, B., Traullé, O., Bouhours, G., Sinardet, B., Barri
é, J., Vinson, J.P., Boutet, F., Berthod, C. and Clémençon, A., 2009. Turbulent
air‐sea fluxes in the Gulf of Guinea during the AMMA experiment. Journal of
Geophysical Research: Oceans, 114(C4).

Brown, T., Mann, B., Ryder, N., Subbiah, M., Kaplan, J.D., Dhariwal, P., Neelakantan,
A., Shyam, P., Sastry, G. and Askell, A., 2020. Language models are few-shot
learners. Advances in neural information processing systems, 33: 1877-1901.

Chen, S., Hu, C., Barnes, B.B., Wanninkhof, R., Cai, W.-J., Barbero, L. and Pierrot, D.,
2019. A machine learning approach to estimate surface ocean pCO2 from
satellite measurements. Remote Sensing of Environment, 228: 203-226.

Cummins, D.P., Guemas, V., Cox, C.J., Gallagher, M.R. and Shupe, M.D., 2023.
Surface Turbulent Fluxes From the MOSAiC Campaign Predicted by Machine
Learning. Geophysical Research Letters, 50(23).

Cutolo, E., Pascual, A., Ruiz, S., Zarokanellos, N.D. and Fablet, R., 2024. CLOINet:
ocean state reconstructions through remote-sensing, in-situ sparse observations
and deep learning. Frontiers in Marine Science, 11.

Ge, L., Wang, G., Huang, B., Cao, C., Chen, X. and Chen, G.

Jumper, J., Evans, R., Pritzel, A., Green, T., Figurnov, M., Ronneberger, O.,
Tunyasuvunakool, K., Bates, R., Zidek, A., Potapenko, A., Bridgland, A., Meyer,
C., Kohl, S.A.A., Ballard, A.J., Cowie, A., Romera-Paredes, B., Nikolov, S.,
Jain, R., Adler, J., Back, T., Petersen, S., Reiman, D., Clancy, E., Zielinski, M.,
Steinegger, M., Pacholska, M., Berghammer, T., Bodenstein, S., Silver, D.,
Vinyals, O., Senior, A.W., Kavukcuoglu, K., Kohli, P. and Hassabis, D., 2021.
Highly accurate protein structure prediction with AlphaFold. Nature, 596(7873):
583-589.

Monin, A.S. and Obukhov, A.M., 1954. Basic laws of turbulent mixing in the surface
layer of the atmosphere. Contrib. Geophys. Inst. Acad. Sci. USSR, 151(163):
e187.

Rasp, S., Pritchard, M.S. and Gentine, P., 2018. Deep learning to represent subgrid
processes in climate models. Proc Natl Acad Sci U S A, 115(39): 9684-9689.

Silver, D., Hubert, T., Schrittwieser, J., Antonoglou, I., Lai, M., Guez, A., Lanctot, M.,
Sifre, L., Kumaran, D. and Graepel, T., 2018. A general reinforcement learning
algorithm that masters chess, shogi, and Go through self-play. Science,
362(6419): 1140-1144.

Yu, L., 2019. Global Air–Sea Fluxes of Heat, Fresh Water, and Momentum: Energy
Budget Closure and Unanswered Questions. Annual Review of Marine Science,
11(1): 227-248.

Zhou, S., Shi, R., Yu, H., Zhang, X., Dai, J., Huang, X. and Xu, F., 2024. A Physical‐
Informed Neural Network for Improving Air‐Sea Turbulent Heat Flux
Parameterization. Journal of Geophysical Research: Atmospheres, 129(17).

---

## Author Comment (AC3)

**Responses to the Comments and Suggestions**

**Reviewer 1:**

**Summary and Merit:**

Global air-sea flux estimates are useful for understanding the transport of heat and water throughout the globe. With this dataset, the authors use a physics-constrained data-driven method to generate a dataset at moderate resolution (0.25 degrees) from 1993-2017. A key improvement is realistic representation of the ratio of SHF to LHF. While I think the work itself is a very interesting exercise and think this has strong potential to be a useful dataset, I do have a significant concern that I would like to see discussed.

Re: Thank you for your commonts. We have carefully considered all your comments and suggestions and made corresponding point-by-point responses and revisions. Specifically, reviewer comments are shown in black, our responses in blue, and the corresponding revisions in the manuscript are highlighted in red. We hope that our responses and the revised manuscript would be satisfactory.

**Main comment:**

I am not entirely convinced that the training dataset has large enough spatial and temporal coverage for the neural network to accurately generalize and produce a product with global-scale coverage. In particular, from Figure 2, it looks like the training observations are disproportionately from the tropical ocean. Outside of the tropics, only the northeast Pacific and North Atlantic appear to have (visually) reasonable coverage. To evaluate performance on "unseen" locations, the authors employ spatial-informed cross validation. While this procedure demonstrates that predictions are reasonably accurate at the different spatial domains that are part of the training set, this does not indicate that predictions will be accurate in regions where there are not any existing data. For instance, there are many locations in the southern hemisphere presumably characterized by different dynamics than the locations in training dataset. The comparisons between basins presented later are also only reflective of the locations in Fig 2, I think. Of additional concern is that there are many variables used in training which likely have a relationship with air-sea fluxes that is very location-specific.

I do appreciate that the authors attempt to address this issue with the above, but I don't think this goes far enough. I also acknowledge that this is not an easy comment to address (i.e., more buoy measurements cannot be used if the buoys do not exist). But, I

still think the discussion of this could be improved. One idea might be to perform an even more targeted form of cross-validation, e.g., removing one of the isolated locations from training to see how well the neural network performs— and use this to quantify uncertainty. E.g., Remove the single location south of Australia from training, and see how the NN performs for predictions of that location when only the others are used in training. The current Figures 3-5 lump data together from different regions, so it is not possible to determine how well performance is for the isolated locations. Such an approach could be repeated for other single isolated locations to get a generalized idea of uncertainty at several of the remote locations not included in training. There probably could be other ways to address it as well. But in any case, there needs to be some manner of disclaimer- the R values and RMSE shown represent performance at the locations used in training and do not necessarily indicate the same performance in a generalized global sense.

Re: We appreciate the reviewer's thoughtful and constructive comments regarding the limitations in spatial coverage of the training dataset and agree that, despite our use of spatial-informed cross validation, the current approach does not fully quantify performance in truly unseen regions. Additionally, we fully agree with the reviewer's concern that the relationships between air-sea fluxes and the selected input variables may be location-specific due to regional dynamics.

In response to the reviewer's suggestion, we conducted an additional targeted cross- validation focusing on isolated locations in the high-latitude Southern Hemisphere.

Specifically, we selected two buoy sites [Southern Ocean Flux buoy from the Upper

Ocean Processes Group (UOP) and Global Southern Ocean Station buoy from the

Ocean Observatories Initiative (OOI)], which are geographically isolated from the rest of the training dataset. We removed the data from each of these locations from the training dataset in turn and evaluated the neural network's performance at those sites.

In addition, we calculated the model's statistical metrics at the two sites under spatially- informed cross-validation and made comparison with the performance under the targeted cross-validation. The resulting metrics help assess the model's extrapolation capability in underrepresented regions. In the revised manuscript, details of this analysis have been added as follows in the sixth paragraph of Section 3.5 and presented in Tables

S4–S7.

"We applied a spatial 10-fold cross-validation, which provides a more generalized assessment than traditional random cross-validation, to evaluate the BrTHF model.

However, it is important to acknowledge that the spatial distribution of the training dataset is inherently imbalanced, with a heavy concentration of observations in the

Tropics and the Northern Hemisphere. In contrast, the Southern Hemisphere—

particularly the Southern Ocean—suffers from sparse or even missing observational coverage. Given that the environmental conditions in these underrepresented or data- sparse regions may differ significantly from those captured in the training dataset, the selected input variables for the observations may lead to large uncertainty in the model's performance in these areas. To further assess the model's ability to extrapolate to such regions, we conducted an additional targeted cross-validation. Specifically, we excluded stations from the Southern Ocean [i.e., Southern Ocean Flux Station (SOFS)

and Global Southern Ocean Station (GSOS)] from the training dataset and used them solely for validation. Results presented in Tables S4 and S5 show that the BrTHF model achieved the best performance in terms of LHF and $\beta$ at the SOFS with lower RMSE

of 15.6 W/m$^2$ and 0.73 and higher values of r of 0.96 and 0.34, respectively, while its

SHF was slightly outperformed by ERA5 and the physics-free NN model. At the GSOS,

BrTHF yielded more accurate estimates for SHF and $\beta$ with RMSEs of 6.38 W/m$^2$ and

0.74 and values of r of 0.95 and 0.16, respectively, compared to other products, while its LHF was marginally less accurate than that of SeaFlux and the physics-free NN

model. Moreover, under both spatially-informed cross-validation and targeted cross- validation, the model demonstrates comparable accuracy at the two sites, as shown in

Figures S4–S7. These findings suggest that BrTHF retains competitive accuracy of SHF,

LHF and $\beta$ even in regions entirely excluded from training, reflecting promising generalization."

Furthermore, we now include a disclaimer in the revised manuscript emphasizing that the reported R values and RMSE reflect model performance only at locations with available observation at the end of the sixth paragraph of Section 3.5. We hope these additions address the reviewer's concerns and improve the clarity of model generalization.

"While these results are encouraging, it is important to note that the validation remains limited to a small number of sites with available observations. Therefore, the reported r values and RMSE reflect model performance in these specific locations and do not necessarily guarantee similar accuracy in broader, unobserved ocean regions."

**Line-by-line comments and suggestions:**

Title/abstract – It might be helpful to explicitly mention that these are bulk flux predictions

Re: Thank you for your suggestion. We have revised the title (Bowen ratio-constrained global dataset of bulk air-sea turbulent heat fluxes from 1993 to 2017) and abstract to explicitly mention that the products are bulk flux predictions.

L66 – typo seriously "imped"

Re: Thank you for your comment. We have revised "imped" to "impeded".

L68 – change "ascribed" to "attributed"

Re: Revised as suggested.

L70-77 – I think this section should be more explicit on what the problems are with existing parameterizations

Re: Thank you for your comment. We have revised and expanded the second paragraph of Section 1 to more explicitly highlight the deficiencies in existing parameterizations.

The revised text is as follows:

"More explicitly, existing parameterizations often rely on simplified assumptions about atmospheric stability and boundary layer dynamics, which may not hold under diverse environmental conditions. For instance, most bulk algorithms are optimized for moderate wind regimes, resulting in degraded performance and increased uncertainty when applied under weak wind regimes (Brunke, 2002; Jiang et al., 2024). At very high wind speeds, however, observations show that the drag coefficient can decrease due to sea spray and whitecap formation, reducing effective surface roughness and potentially biasing flux estimates (Cai et al., 2025). In addition, simplifications in the treatment of sea surface skin temperature, saturation humidity, and air density in the parameterizations can also introduce substantial uncertainty (Brodeau et al., 2017).

Together, these limitations can contribute a lot to the biases in the SHF and LHF

estimates and can even lead to the unphysical estimations of $\beta$, as Wang et al. (2025)

reported."

L78 – clarify what upscaling means in this context

Re: Thank you for your valuable comment. In the revised manuscript, we have clarified what upscaling means in the third paragraph of Section 1 as follows:

"Machine learning techniques have been extensively applied to upscale point-scale in-
situ measurements of a single variable (such as soil moisture, roughness, or temperature)
into grid-scale global datasets (Wang et al., 2023; Peng et al., 2022; O and Orth, 2021;
Nelson et al., 2024; Fu et al., 2023)."

L93 – "patterns"

Re: Thank you for pointing out our typo. We have revised "pattern" to "patterns".

L103 – I don't understand what "their synergistic changes" refers to

Re: Thank you for your comment. We apologize for the lack of clarity in the original manuscript and have revised the sentence as follows:

"To improve the estimation of SHF, LHF, and $\beta$ in a coordinative framework, we recently proposed an innovative Bowen ratio-informed data-driven model by considering the synergistic changes [on the one hand, ensuring physical consistency (i.e., SHF/LHF = $\beta$); on the other hand, achieving high-accuracy estimations of SHF,

LHF, and $\beta$ simultaneously] using a Random Forest (RF) technique (Wang et al., 2024)."

L107 – ambiguous whether "this work" refers to the 2024 work or the present paper

Re: Thank you for your comment. In the revised manuscript, we have specified that

"this work" refers to Wang et al. (2024).

L118 – "three fold"

Re: Thank you for your comments. We have revised "three-folds" to "three fold".

L146-161 – I think these datasets should be listed in table form, not as a long paragraph.

It would make this much easier to read.

Re: Thank you for your suggestion. In the revised manuscript, we have reorganized those datasets into Table 1 to improve clarity and readability.

L202 – By forcing variables, it might be helpful to clarify that this means variables used in training the neural network

Re: Thank you for your valuable comment. By following the suggestions from you and reviewer 2, we have revised the title of Section 2.2.1 from "Forcing datasets" to

"Learning datasets for training the neural network".

L214 – not sure it's necessary to list these out in paragraph form. To be concise it might be better to simply refer to the relevant table.

Re: Thank you for your suggestion. We would like to clarify that the information has already been summaried in the Table 1 in the original manuscript. Following your suggestion, we have removed the detailed dataset descriptions for conciseness.

L276 – I am concerned that the relationships between air sea fluxes and these 11

variables are not globally generalizable.

Re: Thank you for your comment. Please refer to our comprehensive and detailed response to your Main Comment.

L316 – Might be helpful to add a short explanation on why you chose these metrics

Re: Thank you for your suggestion. In the revised manuscript, we have added a brief explanation in the fourth paragraph of Section 2.4 as follows:

"These metrics—BIAS, RMSE, and r—comprehensively evaluate model performance, representing systematic deviation, dispersion between observations and estimates, and the strength and direction of the linear relationship, respectively."

L363-383, Fig 5 – While performance in terms of RMSE is clearly improved as explained, depending on the application it might be considered a deficiency that BrTHF

does not reproduce extreme values of Bowen ratio that we know exist from the observations (i.e. the distribution is not necessarily better represented than the other models). I think this needs to be explicitly discussed.

Re: Thank you for your comment. We acknowledge that our model does not fully capture the extreme values of $\beta$, which is a deficiency to be addressed in future work. However, from Figure 5, we would like to clarify that, although our model predicts $\beta$ within ±2—slightly narrower than the observed range of ±5, other models and products, while capable of reaching ±5, generate numerous $\beta$ values far beyond the observed range (e.g., 5 to 500 or –5 to –500). The distribution of $\beta$ predicted by the BrTHF model is overall relatively better aligned with the observations compared to other products and models.

In short, although the BrTHF model slightly underestimates the extreme values of $\beta$, it avoids the occurrence of unrealistic outliers seen in other products, making it overall better aligned with observations.

In the revised manuscript, we have now explicitly discussed this limitation of $\beta$ in the eighth paragraph of Section 3.5 as follows:

"While incorporating the constraint of $\beta$ into the model effectively suppresses outliers, it also compresses the physically plausible range of $\beta$. As a result, the distribution of $\beta$ shown in Figure 5(i) differs notably from other products and models, which may limit the product's applicability for users interested in extreme $\beta$ values. It is highlighted that although the BrTHF model slightly underestimates the extreme values of $\beta$, it avoids the occurrence of unrealistic outliers (e.g., 5 to 500 or –5 to –500) seen in other products, making it overall better aligned with observations. Moving forward, we aim to enhance the model's ability to preserve physically plausible extremes while maintaining robustness against outliers in future updates.

L400+ - I think it might be useful to compare the performance by basin to the amount of data coverage between basins. This might help explain why the model performed the way it did.

Re: Thank you for your suggestion. As recommended, we evaluated several indicators of the data coverage across ocean basins, including number of buoys, number of samples, buoy density, sample desity, nearest neighbor distance (NND, the distance between a given point and its closest neighboring point) and standard deviation of NND in Table S6. By computing NND for all sample points and then calculating the mean and standard deviation, we can characterize the density and spatial uniformity of the point distribution. In general, a higher mean indicates a sparser distribution, whereas a higher standard deviation reflects greater spatial heterogeneity.

These indicators were then used to represent data coverage across basins and, in combination, to compare model performance among different ocean basins. In the revised manuscript, the relevant findings have been incorporated into the fifth paragraph of Section 3.5 as follows:

"Based on Figure 2 and Table S6, we observe that the spatial coverage of observations varies across different ocean regions: the Northern Hemisphere generally has higher coverage than the Southern Hemisphere, with the Northern Pacific Ocean exhibiting the highest coverage, while the Arctic Ocean shows the lowest. Comparing spatial coverage with accuracy metrics reveals a more complex relationship between model performance and data coverage. Specifically, the values of r tend to be lower in regions with lower coverage — a pattern consistent across SHF, LHF, and $\beta$. However, RMSE does not follow this trend. For SHF and $\beta$, RMSEs in the Northern Hemisphere are generally higher than those in the Southern Hemisphere. Similarly, for LHF, RMSEs are higher in the Northern Hemisphere except in the Indian Ocean, where the pattern differs."

Fig 7 – I would recommend to use a color other than blue for the second and third columns. As is, it is confusing that dark blue = poor performance in column 1, but dark blue = good performance in columns 2 and 3.

I also think it should be very clear that the basins here just represent the buoy locations that are available in those basins; not uniform coverage in them.

Re: Thank you for your suggestion. In the revised manuscript, we updated the color schemes in the second and third columns to a diverging colormap for more consistent interpretation. We also clarified in the caption of the Figure 7 that the displayed ocean basins only reflect the locations of available buoy observations rather than uniform coverage as follows:

"It should be noted that the statistical metrics for each ocean basin were calculated using observations from the available buoys within the corresponding basin."

[Figure]

**Figure 7. Heatmaps of BIAS, RMSE and r metrics for the validation of estimated daily SHF**

**(a - c), LHF (b - e), $\beta$ (f - i) and $\beta$ (-5 ≤ $\beta$ ≤ 5, j - l) from the BrTHF model, the physics-free NN**

**models and the seven products against the in-situ observations across different ocean basins.**

**It should be noted that the statistical metrics for each ocean basin were calculated using**

**observations from the available buoys within the corresponding basin**

L448-449 – That looks true for all datasets, not just BrTHF from Figure 8. I would recommend to clarify.

Re: Thank you for your comment. We agree that the less pronounced peak in SHF and

$\beta$ compared to LHF is observed across all products in Figure 8, not just BrTHF. The sentence has been revised to clarify this seasonal pattern.

Fig 8-9 – Is there a measure of uncertainty in these long-term averages that could be included on the plots?

Re: Thank you for your suggestion. We chose the commonly used standard deviation to represent uncertainty of the long-term averages and have added it to Figures 8 and 9

as follows:

[Figure]

**Figure 8. Intra-annual cycles of area-weighted global monthly mean of SHF (a), LHF (b) and**

**β (c) from the eight products from 1993 to 2017. The shaded areas indicate ±1 standard**

**deviation around the mean.**

[Figure]

**Figure 9. Inter-annual evolution of area-weighted global mean SHF (a - b), LHF (c - d) and $\beta$ (e - f) from 1993 to 2017. The trends were calculated based on the Sen's slope method. The \* in the sub-figures (b, d and f) represent the trend passed the Mann-Kendall significant test (p < 0.05). The shaded areas indicate ±1 standard deviation around the mean.**

L472 – "rest of the products"

Re: Thank you for your suggestion. We have revised "the rest five products" to "rest of the products".

L482-483 – I would recommend to speculate on what regions/mechanism may have caused this positive trend, as it differs from the other products.

Re: Thank you for your comment. As shown in Figure 9, the differences between trends in SHF and LHF from BrTHF product were relatively lower than those from other products. In contrast, except for MERRA2, other products show a stronger increasing trend in LHF than in SHF (e.g., IFREMER, SeaFlux, and ERA5), or an increasing trend in LHF accompanied by a decreasing trend in SHF (e.g., JOFURO3, OAFlux, and OHF). This is likely the cause of the different $\beta$ trend in BrTHF (weakly positive, close to zero, and not statistically significant), and such differences can be further attributed to disparities in the accuracy of SHF, LHF, and $\beta$ among the products. Considering that our validation results indicate higher overall accuracy of BrTHF product, the $\beta$ trend in our product may be reasonable. Nevertheless, the reliability of long-term trends ultimately requires further observational data to determine which product provides the most accurate representation.

In the revised manuscript, we have clarified the possible reason in the third paragraph of Section 3.2 as follows:

"However, the BrTHF product exhibited a weak positive trend, which may be attributed to the relatively smaller differences between the SHF and LHF trends in BrTHF compared to those in other products."

Sec 3.3 – This section implies that performance between BrTHF and Seaflux-ERA5 is similar, even in regard to Bowen ratio which earlier seemed to be the point of significant improvement for BrTHF. Please comment on this.

Re: Thank you for your comment. We would like to clarify that the large-scale spatial patterns of air-sea turbulent heat fluxes are primarily shaped by atmospheric circulation and sea surface properties (e.g., sea surface temperature, and salinity), which result in broadly similar spatial structures across different products as the reviewer pointed out. However, notable differences remain as shown in the difference maps (first and second rows, fourth column) and scatter plots (fourth row, first and second columns) of Figures 10-12. For instance, BrTHF shows significantly higher SHF values in the high-latitude Northern Hemisphere compared to SeaFlux, with greater dispersion in the scatter plots. These spatial and statistical differences reflect the improvements achieved by our model and have been described in Section 3.3 of the original manuscript.

In the revised manuscript, we have added a discussion in Section 3.3, third paragraph, to clarify the potential explanation as follows:

"In addition, the OHF product did not reproduce similar large-scale spatial patterns of air–sea turbulent heat fluxes observed in BrTHF, ERA5, and SeaFlux, which are primarily shaped by atmospheric circulation and sea surface properties (e.g., sea surface temperature and salinity)."

Fig 13 – It's a bit confusing that the labels on the color bar are below the plots on the left. It might be more intuitive to add a title above each subplot rather than a colorbar label.

Re: Thank you for your suggestion. In the revised manuscript, we have moved the labels to the top-left corner of each subplot in Figure 13 to improve readability and make the figure more intuitive.

[Figure]

**Figure 13. Spatial maps of inter-annual trends for SHF (a), LHF (c), and $\beta$ (e) from the BrTHF**

**product for the period 1993 to 2017. The trends were calculated using the Sen's slope method.**

**Dotted areas indicate oceans where the p-value of the Mann-Kendall significance test is less**

**than 0.05. Panels (b), (d) and (f) represent the inter-annual trends of zonal annual averages for SHF, LHF and $\beta$, respectively.**

L553-555 – Do we trust these results, considering that there was significant uncertainty at high latitudes (and the NN was trained on few observations from high latitudes)? Could this be an artifact of the training data/procedure?

Re: Thank you for your comment. Due to the scarcity of long-term observations in high-latitude oceans, we assessed the reliability of simulated trends of BrTHF in these regions by comparing them with the corresponding trends from seven widely used products. As shown in Figures S2–S4, in these high-latitude regions, the trends simulated by the BrTHF are largely consistent with those of most other products—for example, SHF exhibits a pronounced increase in the Kara Sea, Gulf Stream, Baffin Bay, Brazil Current, Sea of Okhotsk, and Sea of Japan, with differences mainly in magnitude. Given that these products are developed based on physically well-founded models, we consider the trends simulated by our product to be reliable.

In the revised manuscript, we have added a discussion about the reliability of simulated trends in the fourth paragraph of Section 3.5 as follows:

"The generalization capability of the model can also affect the accuracy of simulated long-term trends. In Figure 13, we present the spatial distributions of long-term trends for SHF, LHF, and $\beta$ simulated by the BrTHF product. Considering the scarcity of training data in high-latitude oceans, the simulated long-term trends in these regions may be associated with larger uncertainties. However, due to the lack of long-term observations in high-latitude oceans, we cannot validate the simulated trends using observational records as done in previous studies for mid- and low-latitude regions (Tang et al., 2024; Weller et al., 2022). To address this, we examined the spatial distribution of long-term trends from the other seven widely used products. Specifically, in these high-latitude regions, the trends simulated by the BrTHF are largely consistent with those of most other products—for example, SHF exhibits a pronounced increase in the Kara Sea, Gulf Stream, Baffin Bay, Brazil Current, Sea of Okhotsk, and Sea of

Japan, with differences mainly in magnitude."

L588 – "custom"

Re: Thank you for your comment. We have revised "customed" to "custom".

L590 – I'm unconvinced that the absence of outliers is an improvement, since outliers exist in the observations. Please comment on this.

Re: Thank you for your comment. We acknowledge that outliers do exist in observations; however, many of the outliers are likely caused by measurement errors.

Considering that such outliers can severely impede model training and evaluation, we deemed it necessary to constrain the $\beta$ in a reasonable range to enable simultaneous high-accuracy estimation of SHF, LHF, and $\beta$.

Specifically, we calculated the cumulative distribution of daily $\beta$ for each product and their ensemble (across all products). The medians of the 1$^{st}$ and 99$^{th}$ percentiles, approximately -5 and 5, respectively, were selected as the minimum and maximum of valid daily $\beta$, as shown in Figure S1. We did not derive the constraints of $\beta$ directly from observations, primarily because the limited spatial coverage of observations may not provide a range that is generally applicable across all ocean basins. While simulated data offer global representativeness, they may also contain outliers. Therefore, we manually set a reasonable $\beta$ range based on the 1$^{st}$ -99$^{th}$ percentiles (in ascending order), as already presented in the fifth paragraph of Section 2.1. This range provides a robust basis for model development, ensuring that SHF, LHF, and $\beta$ can be jointly estimated with high accuracy.

In the revised manuscript, we have clarified the importance of absence of $\beta$ outliers in the fifth paragraph of Section 2.1 as follows:

"Although outliers exist in observations, some are likely caused by measurement errors.

Considering that such outliers can severely impede model training and evaluation, it was necessary to constrain $\beta$ within a reasonable range to enable simultaneous high- accuracy estimation of SHF, LHF, and $\beta$.”

L609-618 – I'm not sure that this isn't also true for the present dataset based on looking at Figure 2

Re: Thank you for your comment. This issue appears closely related to model generalization and has been discussed in detail in the Main Comment.

L666 – Performance in terms of SHF/LHF did not clearly look superior based on the plots. Please clarify that the largest improvement is in Bowen ratio.

Re: Thank you for your comment. In the revised manuscript, we have clarified that the most significant improvement achieved by the BrTHF model is in the estimation of the

$\beta$, while its performance in estimating SHF and LHF is generally comparable to or slightly better than other models and products in the second paragraph of Section 5 as follows:

"The BrTHF model demonstrated the most significant improvement in estimating the

$\beta$, while its performance in estimating SHF and LHF was generally comparable to or slightly better than that of the physics-free NN models and the seven widely used air- sea turbulent heat products (including the JOFURO3, IFREMER, SeaFlux, ERA5,

MERRA2, OAFlux and OHF products)."

Reference:

Brodeau, L., Barnier, B., Gulev, S.K. and Woods, C., 2017. Climatologically Significant
  Effects of Some Approximations in the Bulk Parameterizations of Turbulent
  Air–Sea Fluxes. Journal of Physical Oceanography, 47(1): 5-28.
Brunke, M.A., 2002. Uncertainties in sea surface turbulent flux algorithms and data sets.
  Journal of Geophysical Research, 107(C10).
Cai, L., Wang, B., Wang, W. and Feng, X., 2025. The Impact of Air–Sea Flux
  Parameterization Methods on Simulating Storm Surges and Ocean Surface
  Currents. Journal of Marine Science and Engineering, 13(3).

Jiang, Y., Li, Y., Lu, Y., Wu, T. and Gao, Z., 2024. Evaluating modifications to air–sea
momentum flux parameterizations under light wind conditions in CAM6.
Climate Dynamics, 62(10): 9687-9701.

Tang, R., Wang, Y., Jiang, Y., Liu, M., Peng, Z., Hu, Y., Huang, L. and Li, Z.-L., 2024.
A review of global products of air-sea turbulent heat flux: accuracy, mean,
variability, and trend. Earth-Science Reviews, 249.

Wang, Y., Tang, R., Liu, M., Huang, L. and Li, Z.-L., 2025. Bowen ratio-constrained
global dataset of air-sea turbulent heat fluxes from 1993 to 2017. Earth System
Science Data Discussions, 2025: 1-41.

Weller, R.A., Lukas, R., Potemra, J., Plueddemann, A.J., Fairall, C. and Bigorre, S.,
2022. Ocean Reference Stations: Long-Term, Open-Ocean Observations of
Surface Meteorology and Air–Sea Fluxes Are Essential Benchmarks. Bulletin
of the American Meteorological Society, 103(8): E1968-E1990.